# Using WHODAS 2.0 to Assess Functional Impairment in People with Depression: Should Employment Receive More Attention?

**DOI:** 10.3390/ijerph18094552

**Published:** 2021-04-25

**Authors:** Yu-Chen Chiang, Tsan-Hon Liou, Hsin-Chien Lee, Reuben Escorpizo

**Affiliations:** 1Department of Psychiatry, Taipei Medical University Shuang Ho Hospital, Taipei 23561, Taiwan; 18481@s.tmu.edu.tw; 2Department of Physical Medicine and Rehabilitation, Taipei Medical University Shuang Ho Hospital, Taipei 23561, Taiwan; 3Department of Physical Medicine and Rehabilitation, School of Medicine, College of Medicine, Taipei Medical University, Taipei 11031, Taiwan; 4Department of Psychiatry, Taipei Medical University Hospital, Taipei 11031, Taiwan; ellalee@tmu.edu.tw; 5Research Center for Artificial Intelligence in Medicine, Taipei Medical University, Taipei 11031, Taiwan; 6Department of Rehabilitation and Movement Science, University of Vermont, Burlington, VT 05452, USA; Reuben.Escorpizo@med.uvm.edu; 7Participation, Integration & Social Epidemiology Group, Swiss Paraplegic Research, 6207 Nottwil, Switzerland

**Keywords:** major depressive disorder, functional impairment, WHODAS 2.0, employment

## Abstract

*Background:* Major depressive disorder (MDD) is a highly prevalent mental disorder which causes public health burden and personal disabilities. In people with mental illness, unemployment is an index character of functional impairment. *Methods:* Using the Taiwan Databank of Persons with Disability (TDPD), we collected the WHO Disability Assessment Schedule 2.0 (WHODAS 2.0) scores for people with MDD-associated disability. We recorded and analyzed the scores of participants during a 3-year period to determine the impact of employment on the trajectory of functional change. Logistic regression was performed to analyze the association between employment and changes in WHODAS 2.0 scores. *Results:* In people with MDD-associated disability, unemployed individuals present a worse function initially compared to employed individuals. After a 3-year period, the employed group showed a significant functional improvement in the domains of cognition, mobility, and participation. In logistic regression, the odds of having functional improvement were twice as high for those who were employed compared with those who were not. *Conclusions:* Higher odds of having functional improvement were noted in participants who stay in employment. Programs and strategies to help people with MDD-associated disability resume work warrant more clinical attention and supportive policies from the government.

## 1. Introduction

Major depressive disorder (MDD) is a highly prevalent mental disorder. The total number of people with depression in 2015 was estimated to exceed 300 million [1], which is equivalent to 4.4% of the world’s population.

Although MDD is defined as an episodic mood disorder, it is often chronic and recurrent [2,3]. Given its chronicity, MDD not only increases the public health burden [4] but also causes disability and decreases the quality of life [5,6,7,8]. The severity and recurrence of depression are significantly correlated with MDD-associated disability [9]. Comorbid psychiatric disorders, personality traits such as neuroticism, and the perceived social support are also determinants of functional disability and social adjustment in major depressive disorder [10]. Even after remission from depression, individuals with MDD experience various residual symptoms, including ongoing low mood, insomnia, anxiety, fatigue, and cognitive deficits. Patients with residual symptoms are at an increased risk of functional and interpersonal impairments [11]. A study on 3849 participants that analyzed epidemiological data from the World Mental Health Survey Initiative in Portugal and characterized the association between disability and common mental disorders reported an odds ratio of 3.49 for individuals with MDD that reported disability [8].

According to the Global Health Estimates 2016 from the World Health Organization (WHO) [12], depressive disorders are the third leading contributor to non-fatal health loss (5.8% years on average lost due to disability). Furthermore, they are responsible for 1.7% of global disability-adjusted life years, which is greater than that of other major mental disorders, such as schizophrenia and bipolar disorder. In the Diagnostic and Statistical Manual of Mental Disorders, 5th edition, the Global Assessment of Functioning was replaced by the WHO Disability Assessment Schedule 2.0 (WHODAS 2.0) for the assessment of functional restriction and impairment associated with mental disorders. In the assessment of disability in clinical settings, employment status is an index character, which is associated with cognitive function, personal achievement, economic independence, interpersonal relationships, and social participation. There is considerable evidence for the negative influences of unemployment on depressive symptoms and vice versa [13,14,15].

There were 1,173,978 people with disabilities in Taiwan at the end of 2018 according to official statistics. They accounted for 4.9% of the total population. Chronic mental health conditions and depressive disorder account for 10.8% (*n* = 127,591) and 2.6% (*n* = 30,299), respectively, of total disabilities in Taiwan. To reduce these disabilities and their burden, it is imperative to identify strategies to facilitate the course of recovery from MDD-associated disability. For example, the Taiwan government provides specific job training programs and referral procedures for people with disabilities, and also encourages companies to preserve certain job opportunities for them and to provide a friendly employment environment.

In this study, we hypothesized that among people with MDD-associated disability in Taiwan, unemployed participants had more serious and sustained disabilities than those who were employed. Additionally, after a period of regular treatment, employed participants were more likely to recover from disability. We aimed to investigate the association between employment and functional recovery in people with MDD, in order to provide an evidence-based framework for a rehabilitation strategy in clinical settings and supporting policies from the government.

## 2. Materials and Methods

### 2.1. Participants

We recruited participants between the working age of 18 and 64 from the Taiwan Databank of Persons with Disability (TDPD) from 11 July 2012, to 31 October 2018. The participants received usual psychiatric treatment for the management of MDD under the national health insurance (NHI) system [16] during our study period. 

TDPD is an anonymous databank, and it belongs to the Ministry of Health and Welfare (MOHW) of the Taiwanese government. The MOHW evaluates disability in people with physical or mental disorders who seek healthcare services from government-authorized hospitals and who require social welfare. This helps maintain the basic rights of people with disabilities, in addition to providing them with protection and ensuring their equal participation in society. In the following evaluation, the eligible candidates are given a “disability identification (card)”, related to their welfare. In most situations, the disability card has to be renewed every 1 to 5 years depending on the judgment of the physicians. The TDPD maintains a record of these cards, and it can be used for the purpose of academic research conducted under the supervision of an institutional review board.

In this study, we enrolled individuals with an MDD diagnosis from the TDPD. All recruited participants were evaluated at least twice, 3 years apart during the study period. Thus, we can compare and analyze two sets of evaluation data for each individual. The MDD diagnosis is according to the International Classification of Diseases (ICD) codes (ICD-9-Clinical Modification (CM) diagnostic code: 296; ICD-10-CM diagnostic code: F32.0-F32.5, F33.0-F33.3, F33.9, and F33.40-F33.42).

We excluded individuals who were diagnosed with bipolar disorder during the study period, had a change in their work status after a 3-year period, i.e., from employed to unemployed or vice versa, had other physical disabilities, or had missing demographic data. Eventually, 4079 participants were included in the study (Figure 1).

### 2.2. Measurement Tools

We collected the WHO Disability Assessment Schedule 2.0 (WHODAS 2.0) scores for the people with MDD-associated disability from the TDPD. In present-day Taiwan, the Chinese traditional version of the WHODAS 2.0 is used to evaluate disability. The evaluation at the hospitals was conducted by one psychiatrist and another experienced professional. The psychiatrist evaluates the mental health condition of the International Classification of Functioning, Disability, and Health (ICF) and the disease codes of the ICD-9-CM and ICD-10-CM. The other professionals (who are social workers, occupational therapists or psychologists) evaluate the environmental categories of the ICF and the Chinese Traditional version of the WHODAS 2.0 questionnaire to evaluate the limitations to daily activity and social participation.

The severity of disability is determined by the Disability Eligibility Determination Scale, which was developed by the government based on the concept and the biopsychosocial model of the ICF in 2012 [17].

The WHODAS 2.0 is a generic assessment instrument for health and disability. The WHODAS 2.0 questionnaire comprises six domains with a five-point Likert-type scale (1 = no difficulty, 2 = mild, 3 = moderate, 4 = severe, and 5 = extreme) to measure the difficulty in performing activities. Domain 1 (cognition) assesses communication and thinking activities, specifically, concentration, problem-solving, learning, and communicating. Domain 2 (mobility) assesses standing, moving around inside the home, getting out of the home, and long-distance walking. Domain 3 (self-care) includes questions on bathing, dressing, eating, and staying alone. Domain 4 (getting along) assesses getting along with other people and difficulties that might be encountered with this due to a health condition. Domain 5 (life activities) assesses difficulties in day-to-day activities, such as household, work, and school activities. Domain 6 (participation) represents how other people and the world around them make it difficult for them to take part in society. In this domain, they do not report on their activity limitations, rather, they report on the restrictions they experience from people, laws, and other features of the world [18]. The questionnaire was self-administered, and a trained specialist recorded the results. Scores for each domain ranged from 0 (least difficulty) to 100 (most difficulty) with higher scores indicating more severe disability (0–4 indicates no difficulty, 5–24 indicates mild difficulty, 25–49 indicates moderate difficulty, 50–95 indicates severe difficulty, and 96–100 indicates extreme difficulty). If a respondent’s condition was inconsistent with the item descriptions in the questionnaire for >30 days, such items were recorded as unrated. According to the WHODAS 2.0 manual, the unrated items are calculated as the mean score of the domain [16]. We excluded the four items on work and school activities from the life activities domain and subsequently calculated the scores of the remaining 32 items. We calculated the sum of the standardized scores for the 32 items for every participant as follows: 100× (total scores of each domain)/92.

The WHODAS is based on the conceptual framework of the ICF. During its development, the WHODAS 2.0 was investigated in numerous countries and translated into several languages. The Chinese version of the WHODAS 2.0 questionnaire has been reported to have satisfactory validity and reliability by several studies [19,20]. A systemic review on 810 studies from 94 countries concluded that the WHODAS 2.0 is suitable for assessing health status and disability in various settings and populations [21].

### 2.3. Data Collection

We recorded the scores in six domains and the summary of the WHODAS 2.0 for each participant at baseline (Year 1) and after a 3-year period (Year 3). We recorded and analyzed the scores of participants with constant work status to determine the impact of employment on the trajectory of functional change during the 3-year period.

We also collected demographic variables from the TDPD, including sex, age, educational attainment, work status, socio-economic status, and urbanization level. Sex, age, educational attainment, urbanization, and the severity of disability were considered covariates to adjust for the possible differences in the evaluation. Age was further categorized into a younger group, between 18 and 44 years, and a middle-aged group, between 45 and 64 years. We classified the socio-economic status based on the family income into general, middle–low, and low categories. We obtained this information from the government records of the social welfare system. The general economic status suggested the absence of any specific identification of middle–low or low income.

### 2.4. Statistical Analysis

We used the Chi-square test to analyze distributions and variations of demographic data. As the scores of WHODAS 2.0 in our participants were not normally distributed, the Wilcoxon signed-rank test was used to compare the scores in Years 1 and 3.

In the stratified analysis, individuals who were employed in both Years 1 and 3 were defined as the employment group. Individuals who were unemployed in both Years 1 and 3 were defined as the unemployment group. The Wilcoxon rank sum test was used to compare the scores in Year 1 in the two groups.

Finally, we conducted logistic regression to obtain the odds ratio of functional improvement after a 3-year period in the employment group. We used the SAS software (SAS Institute Inc., Cary, NC, USA) for all statistical analyses. A *p*-value of <0.05 was considered statistically significant.

## 3. Results

### 3.1. Characteristics of People with MDD-Associated Disability

We enrolled 4079 participants with MDD-associated disability. The majority of them were female (67.4%). Other demographic features were as follows: Age between 45 and 64 (60.6%); living in urban areas (54.5%); unemployment (89.5%); having less than 9 years of education (70.8%); and general economic status (95.8%) (Table 1).

The assessment in Year 1 indicated that the individuals with MDD had a standardized WHODAS 2.0 score of 37.7 points, indicating moderate difficulty. The following are the mean scores for each domain in descending order: Getting along (50.4 points, moderate difficulty); participation (48.8 points, moderate difficulty); life activities (42.2 points, moderate difficulty); cognition (40.3 points, moderate difficulty); mobility (22.1 points, mild difficulty); and self-care (11.3 points, mild difficulty) (Figure 2).

### 3.2. Changes in WHODAS 2.0 Scores after 3 Years in People with MDD-Associated Disability

The scores of WHODAS 2.0 in disabled people with MDD after a 3-year period of treatment were collected from TDPD. Analysis using the Wilcoxon signed rank test indicated significant decreases in the scores for cognition, mobility, and participation, as well as the overall scores (*p* < 0.0001), indicating ameliorated difficulties in the aforementioned domains. However, the decrease in the scores for getting along and life activities was less significant, thus suggesting a relatively unobvious functional improvement in the two domains (*p* < 0.05). Nonetheless, a slightly increased score in the self-care domain indicated worsening of self-care function in the participants (Figure 2).

### 3.3. Stratified Analysis of Employment Status

We conducted a stratified analysis according to the employment status to compare the correlation factors and changes in the scores for each group.

Employment status was divided into employed or unemployed. Among the 4079 participants, only 430 (10.5%) people were employed in both Years 1 and 3. The stratified analysis revealed that male sex (14.4% vs. 8.7%), younger age (15.4% vs. 7.4%), and less severity of disability were correlated with a higher probability of sustained employment (Table 2). There was a lower correlation among employment and education years, economic status, and urbanization level.

The analysis in Year 1 indicated that the unemployed group had significantly higher scores in each of the WHODAS 2.0 domains than the employed group (*p* < 0.05). This indicates that unemployed individuals with MDD had more severe disability than employed individuals, i.e., those with a lesser degree of impairment were more likely to be employed in a cross-sectional analysis (Table 3).

After the 3-year period, a significant decrease in WHODAS 2.0 scores, i.e., functional improvement, was noted in all domains in the unemployed group, except Domain 3 (self-care). There was also a significant improvement in the domains of cognition, mobility, and participation in the employed group (Table 3 and Figure 3).

### 3.4. The Association between Employment and Functional Improvement

We performed a logistic regression analysis to explore the association between employment and functional improvement in people with MDD-associated disability. We defined a decrease in the WHODAS 2.0 scores in Year 3 characteristic of functional improvement.

As mentioned above, we categorized 4079 participants into two groups according to their work status, following which we controlled factors including age, sex, education level, economic status, living area, and severity of disability. The odds ratios of functional improvement in the employment group for the six domains and the summary of WHODAS 2.0 were 2.1, 2.3, 2.6, 2.1, 2.7, 1.8, and 2.0 (*p* < 0.0001), respectively (Figure 4). Thus, following covariate adjustment, the employed participants with MDD were more likely to have a functional improvement than their unemployed counterparts during the 3-year period.

## 4. Discussion

### 4.1. Functional Impairment and Depression

In this study, we collected data from 4079 people with MDD-associated disability from the TDPD and analyzed their demographic characteristics. Using WHODAS 2.0 as an evaluation tool, we accessed the changes in their functional impairment in a 3-year period, which is a crucial aspect in the treatment and follow-up of MDD [22].

According to the WHODAS 2.0 scores in Year 1, getting along, life activities, and participation domains displayed the most severe impairment. After the 3-year period, there were significant improvements in cognition, mobility, participation, and total scores (*p* < 0.001). This, in turn, suggested an amelioration of functional impairment in individuals with MDD who obtained treatment under the NHI system in Taiwan. The system covers all medications, supportive psychotherapy, admission services, and emergency services for people with MDD.

However, the limited improvement in Domain 4 (getting along) indicated a higher prevalence of social function disability in depressed individuals compared to other domains of disability. Furthermore, it suggested that depression creates significant impacts on the interpersonal relationships of patients, such as getting along with their families and intimate others, making new friends, and dealing with strangers. The limited improvement in Domain 5 (life activities) highlighted the association between depression and significant and persistent impairments in occupational and/or school functioning.

Compared with other impairment types, social function deterioration in MDD has received the most attention [23,24]. Social function impairment, which is defined as “an individual’s disability to perform and fulfill normal social roles”, is considered an important sign of depression [23]. Furthermore, social function involves communication and social skills, which directly affects marital status and employability.

This impairment could even remain after recovery from core depressive symptoms [23,24,25,26]. Signs such as social withdrawal, social anhedonia (decreased interest in social interactions), increased sensitivity to rejection, and reduced empathy can diminish social function in patients suffering from depression [24].

Moreover, psychological factors, such as self-focused attention, negative cognition, self-verification, reassurance-seeking behavior, and anhedonia, could influence social behavior and cause social rejection [25].

### 4.2. Employability and Depression

The stratified analyses grouped by work status indicated a better life function in employed people, considering their lower WHODAS 2.0 scores. We observed a functional improvement in both groups, except Domain 3 (self-care) in the unemployed group after the 3-year period. This suggested that employment prevents worsening of the function of self-care in people with MDD.

The basic treatment for MDD might lead to partial functional improvement in either of the groups. The adjusted odds ratios of functional improvement in the employment group indicated an increased likelihood of recovery from disabilities.

According to a previous study, the lifetime prevalence of MDD is 1.2% in Taiwan [27]. The estimated number of people with MDD is approximately 300,000. About 10% of people with MDD in Taiwan have been diagnosed with a disability.

In our study, 431 participants (9.7%) were employed in both Years 1 and 3, which revealed the lower employability among people with MDD-associated disability. According to the statistical data from the Taiwan government, 66.4% of the general population aged from 15 to 64 years in 2018 were employed. The employment rate in our original data before we excluded those who had a work status change was 16% (Figure 1). The result echoed the data in OECD [28]: People with mental health problems were 30% to 50% less likely to be employed than those with other health problems or disabilities. Since our study population was limited to those who had records in the databank of disabled people, and we excluded those with a work status change for the purpose of specifying the role of employment, the relatively low employment rate in our study is due to the limitations in the selection process. Nevertheless, the results still represent the fact that most MDD people with disability in Taiwan are unemployed.

Both cross-sectional and longitudinal studies have described the vulnerability to unemployment among the depressive population [29]. A population-based, cross-sectional study in the United States reported an unemployment rate of almost 50% among individuals with depression [30]. Two other population-based longitudinal studies compared initially employed individuals with depression with similarly employed healthy controls and reported that depression was associated with a 20% and 40% greater likelihood of unemployment, respectively [31,32].

Social interaction difficulties are correlated with unemployment and decreased work performance among people with depression [33]. The lower employability among people with depression is not only due to the disease itself but also due to the substantial impairment of life, cognitive, and social function. It was reported that 77.1% of workers lost productive time at work due to health-related problems [33]. Further, depressive symptoms diminish a worker’s quality of life and achievements in the workplace [13,29,30,34]. Depression severity is also an influential factor. A longitudinal study on the depression course over 23 years reported that individuals with more severe depression were less likely to be employed than those with less severe depression [13]. Moreover, our findings indicated that employed individuals with depression were more likely to be younger, male, and have less disability, which is consistent with the findings of the study that used the National Health Interview Survey Disability Supplement Databank in the United States [30].

A systematic review reported that employment had a protective effect on depression [35]. Not only are employed individuals likely to preserve a better life function, but they also have less severe depression trajectories and fewer other health problems [13]. Our findings suggest that employment status can be used to evaluate the functional restriction of depressive people at an initial assessment and during a regular treatment. Prolonged unemployment might be indicative of a more serious disability.

Furthermore, higher occupational prestige, greater work environment resources, and lower work environment stressors have also been reported to have a protective effect against more severe, intractable depression over time and might bolster function [13].

In contrast, a stressful work environment is considered a risk factor for developing depression. Employees who reported a lack of decision latitude, job strain, and bullying were reported to experience increased depressive symptoms over time [36].

In summary, depression can be a predisposing factor of unemployment due to disease symptoms or disability. A work environment lacking enough supportive resources could decrease the motivation and probability of returning to work. Sustained employment can facilitate functional improvement in people with MDD. To minimize adverse outcomes, restoring the ability to work and helping depressed individuals enter and remain in work are noteworthy issues.

### 4.3. Strengths and Limitations

To our knowledge, this is the first study to investigate work status and its association with the functional change in people with MDD-associated disability in Taiwan. Additionally, there are few articles that have discussed similar issues in other countries. The application of WHODAS 2.0 to assess disability and the sample size also strengthened our study.

Our study still had several limitations. First, our study only represented a small proportion of people with depression in Taiwan as the study population was limited to those with MDD-associated disability in the TDPD. This reduced the representativeness of our findings. Second, each individual in our study received general psychiatric treatment with NHI coverage. However, the TDPD did not contain information on the regularity, compliance, and other details of the treatment that have an impact on the functional improvement of people with MDD. Third, the absence of data on the severity and characteristics of depressive symptoms might have had a significant influence on their life function. Fourth, we did not include marital status, a potential confounding factor among the variables, given the absence of this information in the TDPD.

Lastly, we acknowledged the complexity and variance in functional improvement in people with MDD. We should carefully consider each of the suggestions on employment in rehabilitation programs. Future studies should focus on other characteristics and factors associated with functional improvement in people with MDD-associated disability.

## 5. Conclusions

Using the WHODAS 2.0 as the measurement tool, we found a moderate difficulty in the life function in individuals with MDD, and the most significant disabled domain was getting along with others. Employed depressive people had a better function than those who were unemployed. Furthermore, the odds ratios of functional improvement in the employment group in the 3-year treatment period suggest that employment is a considerable facilitating factor of functional improvement in people with MDD-associated disability.

For clinical application, enhancing adherence to treatment and providing individualized rehabilitation programs, such as psychotherapy and supported employment, are substantial strategies to help people with depression recover from disability. Vocational training for people with disabilities can be suggested to selected individuals. The policies to provide a friendly workplace environment and help people with MDD stay in a stable employment warrant more clinical attention and support from the government.

## Figures and Tables

**Figure 1 ijerph-18-04552-f001:**
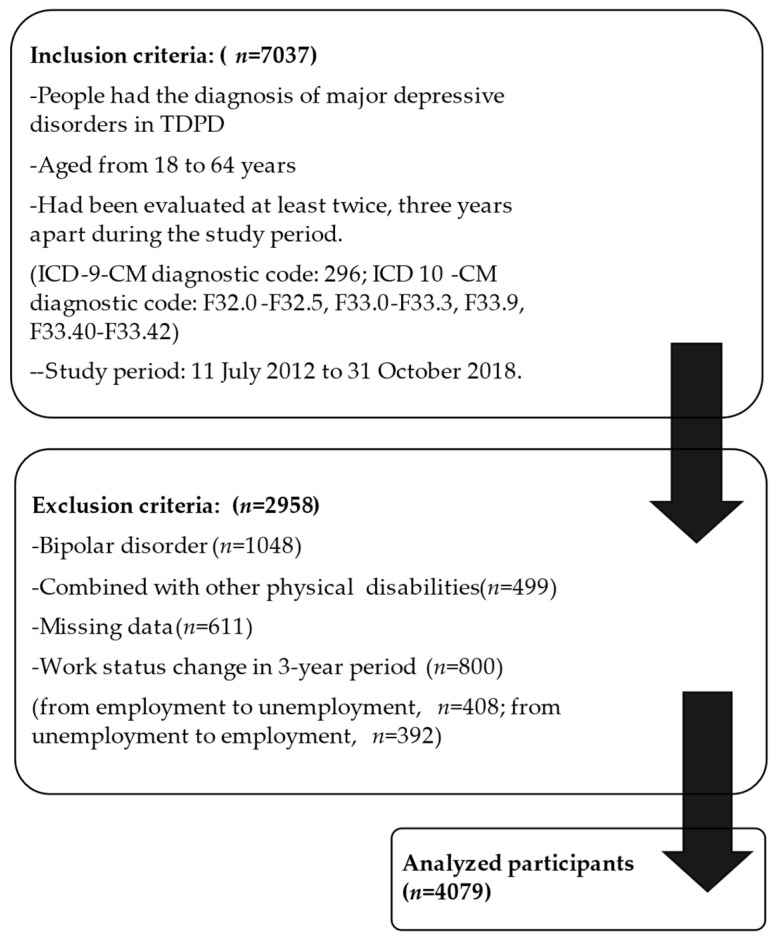
The flowchart of data selection.

**Figure 2 ijerph-18-04552-f002:**
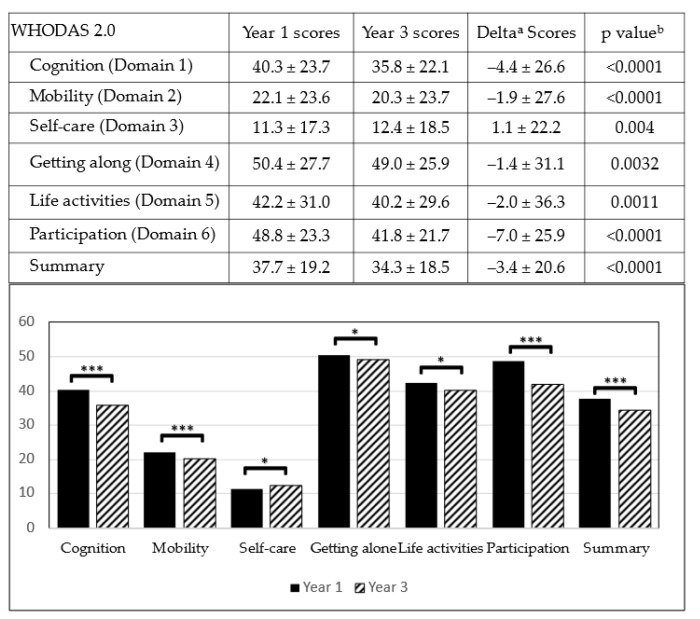
WHODAS 2.0 scores of people with MDD-associated disability in Year 1 and Year 3. (a: Delta = the scores change between Year 1 and Year 3; b: Wilcoxon signed rank test comparing the scores of Year 1 and Year 3; * *p* < 0.05; *** *p* < 0.0001).

**Figure 3 ijerph-18-04552-f003:**
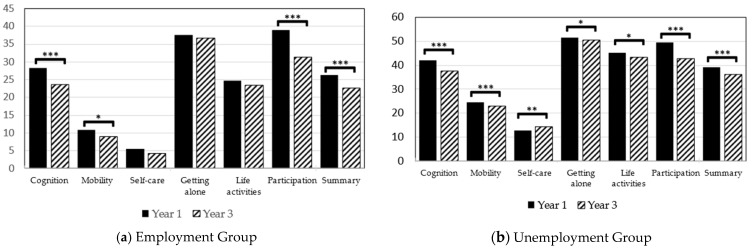
The Changes of WHODAS 2.0 scores in Employment and unemployment group (* *p*<0.05; ** *p*<0.001; *** *p*<0.0001).

**Figure 4 ijerph-18-04552-f004:**
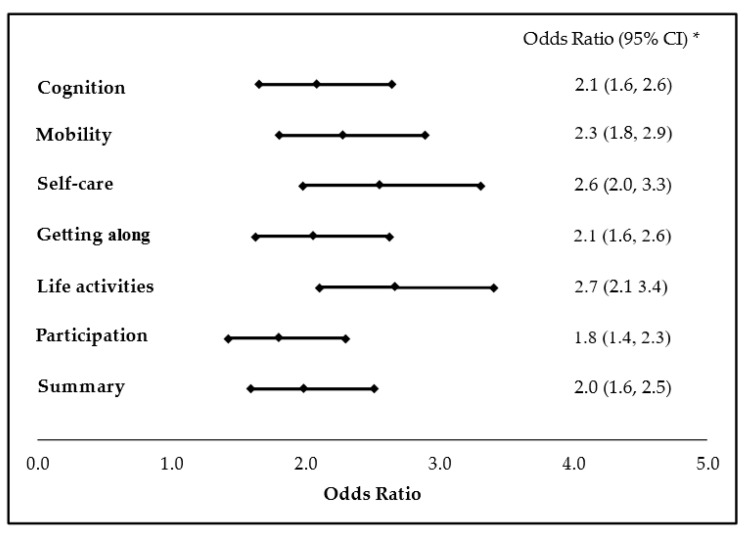
Multivariate logistic regression analysis with adjusted odds ratios for functional improvement in MDD people who are employed. (* The adjusted odds ratios and 95% confidence intervals represent the odds of functional improvement in employed MDD people after adjusting for the covariates including sex, age, education years, family economic status, urbanization level and severity of disability. Decrease of the scores of WHODAS 2.0 in Year 3 is defined as functional improvement).

**Table 1 ijerph-18-04552-t001:** Sociodemographic characteristics of people with MDD-associated disability in Taiwan.

Variables	Male (No., Col%, Row%) (*n* = 1328, 32.5%)	Female (No., Col%, Row%) (*n* = 2751, 67.4%)	*p*-Value ^a^	Total Number (No., %)
**Age (years)**			0.0392	
18-44	554 (41.7%, 34.4%)	1055 (38.4%, 65.6%)		1609 (39.5%)
45-64	774 (58.3%, 31.3%)	1696 (61.7%, 68.7%)		2470 (60.6%)
Mean ± SD	46.1 (± 11.4)	47.5 (± 9.9)	0.0097 ^b^	47.0 (± 10.4)
**Education years**			0.0246	
Less than 9	909 (68.5%, 31.5%)	1977 (71.9%, 68.5%)		2886 (70.8%)
9 and above	419 (31.6%, 35.1%)	774 (28.1%, 64.9%)		1193 (29.3%)
Work Status			<0.0001	
Employment	191 (14.4%, 44.4%)	239 (8.7%, 55.6%)		430 (10.5%)
Unemployment	1137 (85.6%, 31.2%)	2512 (91.3%, 68.8%)		3649 (89.5%)
**Family Economic Status**			0.087	
General	1282 (96.5%, 32.8%)	2624 (95.4%, 67.2%)		3906 (95.8%)
Middle low & Low	46 (3.5%, 26.6%)	127 (4.6%, 73.4%)		173 (4.2%)
**Urbanization level**			0.6353	
Suburban & Rural	611 (46.0%, 32.9%)	1244 (45.2%, 67.1%)		1855 (45.5%)
Urban	717 (54.0%, 32.2%)	1507 (54.8%, 67.8%)		2224 (54.5%)
**Severity of disability ^c^**			0.1097	
Mild	732 (55.1%, 31.2%)	1611 (58.6%, 68.8%)		2343 (57.4%)
Moderate	488 (36.8%, 34.5%)	927 (33.7%, 65.5%)		1415 (34.7%)
Severe & Profound	108 (8.1%, 33.6%)	213 (7.7%, 66.4%)		321 (7.9%)

^a^: Chi-square test; ^b^: Wilcoxon rank sum test; ^c^: Determined by the Disability Eligibility Determination Scale 2012 in Taiwan.

**Table 2 ijerph-18-04552-t002:** The characteristics of employment and unemployment group.

Variables	Employment (No., Col%, Row%) (*n* = 430, 10.5%)	Unemployment (No., Col%, Row%) (*n* = 3649, 89.5%)	*p* value ^a^
**Sex**			<0.0001
Male	191 (44.4%, 14.4%)	1137 (31.2%, 85.6%)	
Female	239 (55.6%, 8.7%)	2512 (68.8%, 91.3%)	
**Age (years)**			<0.0001
18–44	247 (57.4%, 15.4%)	1362 (37.3%, 84.7%)	
45–64	183 (42.6%, 7.4%)	2287 (62.7%, 92.6%)	
Total (mean ± SD)	42.8 (± 8.6)	47.5 (± 10.5)	<0.0001 ^b^
**Education years**			0.1106
Less than 9	290 (67.4%, 10.1%)	2596 (71.1%, 90.0%)	
9 and above	140 (32.6%, 11.7%)	1053 (28.9%, 88.3%)	
**Family Economic Status**			0.1146
General	418 (97.2%, 10.7%)	3488 (95.6%, 89.3%)	
Middle low and low	12 (2.8%, 6.9%)	161 (4.4%, 93.1%)	
**Urbanization level**			0.1988
Suburban and Rural	183 (42.6%, 9.9%)	1672 (45.8%, 90.1%)	
Urban	247 (57.4%, 11.1%)	1977 (54.2%, 88.9%)	
**Severity of disability**			<0.0001
Mild	297 (69.1%, 12.7%)	2046 (56.1%, 87.3%)	
Moderate	116 (27.0%, 8.2%)	1299 (35.6%, 91.8%)	
Severe and Profound	17 (4.0%, 5.3%)	304 (8.3%, 94.7%)	

^a^: Chi-Square test; ^b^: Wilcoxon rank-sum test.

**Table 3 ijerph-18-04552-t003:** The WHODAS 2.0 scores in Year 1 and Year 3, grouped by work status.

	Employment Group (Mean ± SD)			Unemployment Group (Mean ± SD)		
**WHODAS 2.0**	Year 1	Year 3	*Delta*^c,^*	Year 1	Year 3	*Delta*
**Cognition**	28.4 ± 20.0	23.6 ± 18.3 ^b^	−*4.9* ± *22.3*	41.6 ± 23.7 ^a^	37.3 ± 22.0 ^b^	−*4.4* ± *27.1* ^c^
**Mobility**	10.9 ± 16.1	9.0 ± 15.6 ^b^	−*1.9* ± *19.0*	23.5 ± 24.0 ^a^	21.6 ± 24.1 ^b^	−*1.8* ± *28.4* ^c^
**Self-care**	5.5 ± 10.7	4.3 ± 9.0	−*1.2* ± *12.8*	12.0 ± 17.8 ^a^	13.4 ± 19.1 ^b^	*1.4* ± *23.0* ^c^
**Getting along**	37.6 ± 25.7	36.8 ± 24.6	−*0.8* ± *28.1*	51.9 ± 27.6 ^a^	50.4 ± 25.7 ^b^	−*1.5* ± *31.4*
**Life activities**	24.7 ± 26.0	23.4 ± 24.6	−*1.3* ± *31.7*	44.2 ± 30.9 ^a^	42.2 ± 29.5 ^b^	−*2.1* ± *36.9* ^c^
**Participation**	38.9 ± 22.4	31.3 ± 19.5 ^b^	−*7.6* ± *23.4*	50.0 ± 23.1 ^a^	43.1 ± 21.6 ^b^	−*6.9 ±* *26.2*
**Summary**	26.4 ± 16.0	22.7 ± 14.8 ^b^	−*3.7* ± *16.5*	39.0 ± 19.1 ^a^	35.7 ± 18.4 ^b^	−*3.3* ± *21.0*

^a^: Wilcoxon rank sum test *p* value < 0.05, testing the scores of Year 1 in two groups. ^b^: Wilcoxon signed-rank test *p* value < 0.05, testing the scores of Year 1 and Year 3 in the same group; ^c^: Wilcoxon rank sum test *p* value < 0.05, testing Delta in two groups; * *Delta* = the score change between Year 1 and Year 3 in the same group.

## Data Availability

Data is contained within the article.

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
