# Peer review of "Using WHODAS 2.0 to Assess Functional Impairment in People with Depression: Should Employment Receive More Attention?"

_ijerph, 2021, doi:10.3390/ijerph18094552_

Round 1
Reviewer 1 Report
To Authors,
Thank you for your work.
However, there are some problems with statistical analysis.
For example, in the work status, 191 males and 240 females are employed, 31.3% of males and 68.7% of females are calculated as more women working (table 1).
However, due to population differences, it should be calculated as 191 of 1429 males (13.4%) and 240 of 3002 females (8.0%). This result is interpreted contrary to the results presented in this study.
Statistical analysis was performed in the same way in both Tables 1 and 2.
I suggest you do a statistical analysis revise.
Author Response
Dear reviewer:
Thank you for your comments and suggestions that allowed us to greatly improve the quality of the manuscript. We agree with your comments, and we revise the statistical analysis.
The data in Table 1, Table 2 and manuscript are renewed accordingly.
Reviewer 2 Report
I want to thank the authors for the good article. I really don't have a lot to remark on it; though I really tried to find ways to improve your paper, I think it is actually find as it is. Not a criticism, but a question: why the use of non-parametric methods and not a sample t-test? Your sample seems large enough to warrant normality and linearity. I would suggest a more detailed statistical analysis section in the method. The graphs: they seem vague, as if screenshotted and then put into the manuscript. I would suggest you find higher resolutions of the graphs and figures, because they are sometimes hard to read. Also, delete the titles at the top of the graph, because the titles are already below the graphs. The limitations: I don't really understand why you feel that you have a "small sample". The beginning of the discussion could be organised somewhat better. "Functional impairment is a crucial aspect in the diagnosis and follow-up of MDD." - it is a strange way to open your discussion, because, okay, why do you mention this here? I would start with the second part, namely on what your study actually did, then the results. The part "To our knowledge this is the first study to investigate the work status and functional change of people with MDD-associated disability in Taiwan" should be moved to another section. I don't know if the journal allows it, but normally, there is a section "limitations", but coupled with that also the strengths of the study. And I really think you should add that, because your study is, from the albeit limited knowledge I have in that field, quite innovative. Furthermore, your study has a lot more strengths than just being the first. Other than that, I actually don't have any remarks.Author Response
Dear reviewer:
Thank you for your helpful comments and for taking the time to point out options to improve our manuscript. We have revised the manuscript following your suggestions. In the following, we reply to your comments point-by-point.
- “Why the use of non-parametric methods and not a sample t-test? Your sample seems large enough to warrant normality and linearity. I would suggest a more detailed statistical analysis section in the method. ”
Response:
Due to the scores of WHODAS 2.0 in our participants are not normally distributed, we chose the non-parametric methods of Wilcoxon signed-rank test to compare the scores in Year1 and Year3 and Wilcoxon rank sum test to compare the scores in Year1 in two groups. We also add the description in the revised manuscript.(Line 171-173)
- “The graphs: they seem vague, as if screenshotted and then put into the manuscript. I would suggest you find higher resolutions of the graphs and figures, because they are sometimes hard to read. Also, delete the titles at the top of the graph, because the titles are already below the graphs.”
Response:
Thanks for your good advice. We have revised all the graphs in the manuscript.
- “The limitations: I don't really understand why you feel that you have a "small sample".
Response:
Thank you for pointing out the error. It has been corrected as “our study only represent a small part of depression people in Taiwan due to the study population was limited to those with MDD-associated disability in the TDPD.” in the revised manuscript. (Line 353-355)
- “The beginning of the discussion could be organised somewhat better. "Functional impairment is a crucial aspect in the diagnosis and follow-up of MDD." - it is a strange way to open your discussion, because, okay, why do you mention this here? I would start with the second part, namely on what your study actually did, then the results.”
Response:
Thanks for the suggestion. We’ve modified the manuscript accordingly.(Line 253-257)
- “The part "To our knowledge this is the first study to investigate the work status and functional change of people with MDD-associated disability in Taiwan" should be moved to another section. I don't know if the journal allows it, but normally, there is a section "limitations", but coupled with that also the strengths of the study. “
Response:
Thanks for your insightful suggestion. We’ve modified the section “limitation” and add the contents about the strengths of our study. (Line 349-352)
Reviewer 3 Report
The reviewed manuscript tries to discuss the ways of recovery after major depression disorder (MDD). Hence its fits in with the subject of health promotion and seems to be a very important topic for professionals and patients. However, it is not a new idea that work may lead to health impairment as well as may be a form of rehabilitation that helps to return to normality after mental disorder. What is more many prior studies have discussed the associations between being disabled and unemployed and its relations to depression episodes. For this reason the article lacks of information about the originality of the studies. On the other hand an unquestionable advantage of the study is the longitudinal model and relatively large sample of participants with MDD, so it is worth to consider its publication.
Nevertheless some improvements should also be made
- in introduction e.g,.
- it lacks information about the forms of help that is offered to people with MDD, although in the discussion you mentioned about NHI system
- The term “had more obvious” (line 65) is not unequivocal and how you made an operationalziation of it during the study
- The reasons for choosing WHODAS 2.0 tool to measure functioning of participants
- It would be good to add reliability of the WHODAS 2.0 from this study
- In all manuscript p – value should start from small letter (at the moment it is P);
- On the basis of what the authors made an age sub-groups – why you include into one group 18 years adolescents and 44 years old adults, what is more why you include into one age group working adults and old adults 90 years people – for me it is a serious methodological error
- one demographic characteristic is unclear – usually people aged over 65 are retired, the authors included into the sample people aged 90 so their changed in employment during past 3 years probably did not changed at all. In my opinion they should included into the sample only people in working age; the same argument is for youths aged 18
- Figure 3 is a blurry graphics and should be improved
- Line 253 – please add citation
- Line 261 misspelling: “motility,” should be corrected on „mobility”
- Line 266 – 267 “the limited improvement in domain 4 (getting along) indicated a higher prevalence of social function disability in depressed individuals” you did not compare patients with MDD and other groups of disabled people so in my opinion it this conclusion is incorrect
Round 2
Reviewer 1 Report
Thank you for your revision.Author Response
Thank you very much for your assistance.
Reviewer 3 Report
The authors introduced all the amendments I suggest, so in my opinion the text meets the basic publication requirements.
Author Response
Thank you very much.